# Data Descriptions from Large Language Models with Influence Estimation

## Abstract

Deep learning models have been successful in many areas but understanding their behaviors still remains a black-box. Most prior explainable AI (XAI) approaches have focused on interpreting and explaining how models make predictions. But in contrast, we take a different approach via the lens of data because data is one of the most important factors in the success of deep learning models. We would like to understand how data can be explained with deep learning model training via one of the most common media – language. Therefore, we propose a novel approach to understand and extract which information can explain each class inside the dataset well by incorporating knowledge from existing external knowledge bases extracted through large language models such as GPT-3.5. However, the extracted data descriptions may still include irrelevant information, so we propose to exploit influence estimation to choose the most informative textual descriptions, along with the CLIP score. The presented textual descriptions may provide insight into what the trained model focuses on and utilizes for making the prediction. Furthermore, by utilizing recent vision-language contrastive learning as it may provide cross-modal transferability, we propose a novel benchmark task of cross-modal transfer classification to examine the effectiveness of the data description. In experiments with nine image classification datasets, the extracted text descriptions further boost the performance of the trained model with only images. Therefore, it demonstrates that the proposed approach provides information that can explain the characteristics of each dataset that helps the model to train. Through this, we may have insight and inherent interpretability of the decision process from the model. In addition, we show that our approach may help to solve model bias in text-to-image generation tasks.

## 1 Introduction

Deep learning models have successfully applied to various fields and achieved high performances (Huang et al., 2017; Vaswani et al., 2017; He et al., 2016; Dosovitskiy et al., 2020). Despite the rapid advance of performances, understanding their behaviors still remains a black-box. While most prior explainable AI (XAI) approaches have focused on interpreting and explaining how models make prediction, there are few attempts to explain the data. We would like to understand data since the data is one of the most important elements of successful deep learning performance. Among deep learning models, vision models have been quite successful and applied to many applications (Huang et al., 2017; He et al., 2016; Dosovitskiy et al., 2020). Therefore, we try to explain image classes with *human-interpretable* language. For example, in the case of frog class: smooth, moist skin with coloration ranging from green to brown, often featuring various patterns and markings; webbed feet;

To reduce the cost of writing all features of each image class, we use large language models (LLMs) such as GPT-3.5 (Brown et al., 2020). LLMs can be used to understand data and model behavior because they incorporate knowledge about language and thus world. Moreover, we do not simply query LLMs to answer the features of the class, but generate questions using 2-stage prompts with Wikipedia url to utilize external knowledge bases as LLMs may still lack detailed knowledge.

A key advantage of our approach is that we can determine which text is important among the various texts extracted through LLMs. We use influence score and CLIP score to determine the most

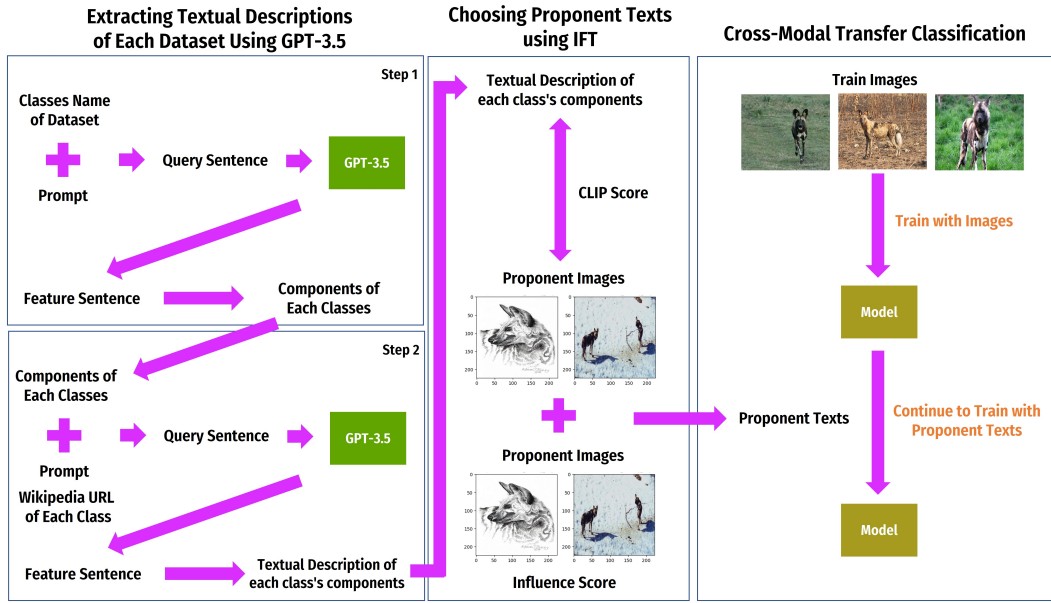

Figure 1: Overview of our framework. Extracting the textual descriptions from GPT-3.5 using predefined prompt process consists of two steps. First step is to get the components of each classes from the raw class name and second step is to get the descriptions of each classes using the components of each classes while providing Wikipedia url. From the extracted textual descriptions, we get the proponent texts using IFT. By using these *proponent texts*, we do cross-modal transfer training the model which is trained with images.

informative textural description for image class. Influence score (Koh & Liang, 2017; Pruthi et al., 2020) is a method that can calculate whether each train image has a positive or negative effect when predicting a test sample and CLIP score is to measure the similarity between the image and the text. In contrast to most prior researches using influence score only on images, we adjust it to the text to find the helpful text to the model by further incorporating CLIP score. We name the helpful text as *proponent text* and the score for determining the proponent text as *influence scores for texts* (IFT).

Recently there has been prior research (Menon & Vondrick, 2023) that uses language to help vision model proposes zero-shot classification using text descriptions extracted through LLMs. In the text description extracted through Menon & Vondrick (2023), they include information that are not helpful visual cues for models to classify images and in some cases, and also the same text is included repeatedly. Through IFT, we solve these problems by filtering out useless information and leaving only important parts that can help the model to classify the image well.

Furthermore, based on the phenomenon of cross-modal transferability (Zhang et al.) in which text input can be used instead of image input for vision models trained only with images, we continue to train the model that are trained with images with proponent texts by giving weight according to the IFT. We call this as 'cross-modal transfer classification'. Compared to training only with images, it shows higher performance when we do cross-modal transfer training the model with proponent texts about nine datasets. Moreover, we show that our approach is not only quantitatively better by achieving higher performance than baseline method (Menon & Vondrick, 2023), but also qualitatively better through human evaluation. The overview of our framework is in Figure 1.

In sum, our contributions are as follows:

- We propose a novel approach to explain data through textual descriptions of image classes from LLM by using 2-stage prompting with the external knowledge base. We can explain image classes through language, which is easier for humans to understand.

- By using the proposed IFT defined as the sum of the influence score and the CLIP score, we can identify which text is the most helpful for explaining each image class. Proponent texts selected through IFT contain only information that helps the model to classify the image

well except unnecessary information. Furthermore, we may get an inherent interpretability of where the black-box model focuses when training.

- We propose cross-modal transfer classification as a novel benchmark task based on the cross-modal transferability phenomenon. In the extensive experiments with nine datasets, we demonstrate that the proponent texts extracted by our proposed approach are textual descriptions that are informative to explain each image class.

## 2  RELATED WORK

Not only achieving high accuracy, but many people also focus on why this model make such decisions. Influence function is one of powerful tool that can explain model decisions. One of the most commonly used idea of influence function (Koh & Liang, 2017) is to see the change of model parameters when perturbing the input data. It uses second-order optimization techniques to efficiently approximate influence score. TracIn (Pruthi et al., 2020) calculates the influence of training data on the loss of a test sample. This computes the influence score by tracing how loss changes when a fixed test sample is given to the training process. In addition to influence functions, there have been various studies related to the explainability of models. GradCAM (Selvaraju et al., 2017) shows which part of the input image is important for the model to make decisions in the form of a heat map. By further expansion, Chefer et al. (2021) provides explainability for prediction of any transformer-based architectures. In this paper, we use influence estimation to have text descriptions of data that are informative and help training of models.

Recently there have been attempts to give explainability of models using large language models (LLMs). Language Guided Bottlenecks (LaBo) (Yang et al., 2023) is an extended model of Concept Bottleneck Models (Koh et al., 2020) that queries LLMs to collect concepts. Menon & Vondrick (2023) proposes alternative zero-shot classification method named 'classification by description'. Menon & Vondrick (2023) uses descriptions extracted through LLMs not just using raw class name. Since Menon & Vondrick (2023) may be the most related work to our approach, we use this as our baseline. Also, Menon & Vondrick (2023) presents some limitations of their method. In the text description extracted through Menon & Vondrick (2023), there is information that are not useful visual cues for the vision-language model and in some cases, the same text is included repeatedly.

One of popular vision-language model, CLIP (Radford et al., 2021) is pretrained with 400 million image-text pairs collected from the internet. This shows high performance in zero-shot setting. Models have trained with vision-language contrastive learning as well as CLIP may provide cross-modal transferability phenomenon (Zhang et al.). Using influence score and CLIP socre, we define *influence scores for texts* (IFT). This serves as a score metric for texts that determines which text is important. We also propose cross-modal transfer classification as a novel benchmark task to prove the effectiveness of textual descriptions of each image classes.

Through the proposed IFT defined as the sum of the influence score and the CLIP score, our proponent texts contain only information that helps the model to classify the image well except for unnecessary information. Moreover, we can utilize external knowledge bases using 2-stage prompting of GPT-3.5 with Wikipedia url. Through the extracted texts, we can not only understand each image class through language, but also use it for training along with images for cross-modal transfer classification.

## 3  APPROACH

As we describe in Figure 1, our framework consists of three main steps. In this section, we describe each step in detail.

### 3.1  EXTRACTING TEXTUAL DESCRIPTIONS USING GPT-3.5

**Extracting Components of Each Class** From the class names of the dataset, we can easily make a question by putting the class names in pre-defined prompt. If the dataset consists of subcategories belonging to one super class, we ask the components of that super class to LLMs. For the case of CUB 200 2011 (Wah et al., 2011) dataset, it has 200 subcategories belonging to birds. In that case,

we ask the components of "birds" which is not the subcategory but the class name. Actually, if we ask for components of super class or components of class name, we can get similar results. However, we did this to provide coherent textual descriptions across all class labels.

This is the form of the question we ask when extracting components of each class:

Q : Can you tell me the components of {class name} from the perspective of appearance?

A :

Then the extracted components are ultimately utilized to extract the textual descriptions of each image class. For example, we can get the components of African hunting dog, one of the classes of Miniimagenet (Vinyals et al., 2016) dataset: body build; coat color; ears; head; eyes. An illustration of this process of extracting components of each class is in Figure 7 in Appendix A. 6 **Extracting Textual Descriptions of Each Class** If we know the components of each class, we can ask the appearance of each component to GPT-3.5. Since GPT-3.5 may still lacked detailed knowledge, we utilize external knowledge bases by providing Wikipedia url of corresponding class. We ask GPT-3.5 to summarize the relevant information on Wikipedia url in one line composed of nouns. If there is no information for what we asked, we ask to find and summarize relevant information among the information that GPT-3.5 knows. This is the form of the question we ask when extracting textual descriptions of each class:

Q : Please summarize the information of appearance about {components of each class} in this {wikipedia url} in one line composed of nouns.

If you couldn't find related information, you must answer general information you know.

A :

A detailed figure that illustrates how we extract textual descriptions of each class using the components of each class is in Figure 8 in Appendix A. And for more information on our 2-stage prompting with Wikipedia url, refer to Appendix A.

## 3.2 CHOOSING PROPONENT TEXTS

We use TracIn (Pruthi et al., 2020) to calculate influence scores for images. Based on influence scores, we can find the proponent images from training data which are helpful for predicting each image of the validation set. Then, we calculate CLIP score between proponent images and corresponding textual descriptions of validation set extracted through GPT-3.5 (described in Section 3.1). Since the influence score is score for images, and the CLIP score is a similarity score between images and texts, we may think that if we fuse these two properly, we could determine the score metric for texts. So, with the sum of influence score for proponent images and CLIP score between images and extracted textual descriptions, we can calculate *influence scores for texts* (IFT). IFT can be defined as:

$$Influence\ score\ For\ Texts = Influence\ Score(I^{train}, I^{val}) + CLIP\ Score(I^{train}, T_c^{val})$$

$$Influence\ Score(I^{train}, I^{val}) = \sum_{i=1}^{k}(\sum_{j=1}^{n_{I^{train}}} \eta_i \nabla loss(w_{t_i}, I_j^{train}) \cdot \sum_{m=1}^{n_{I^{val}}} \nabla loss(w_{t_i}, I_m^{val})) \quad (1)$$

$$CLIP\ Score(I^{train}, T_c^{val}) = max(100 \times cos(E_{I^{train}}, E_{T_c^{val}}), 0) \quad (2)$$

where $I^{train}$ are train image samples and $I_{val}$ are validation image samples. $n_{I^{train}}$ is the size of train dataset and $n_{I^{val}}$ is the size of validation dataset. $T_c$ are extracted textual descriptions of image class $c$. In equation 1, influence score calculates the influence of specific training examples on a given validation sample. The loss of model parameterized by $w$ on training image sample $I$ can be denoted as $loss(w, I)$. Since it is not practical when we consider only one training sample at a time, TracIn (Pruthi et al., 2020) introduces practical influence function via $k$ checkpoints $(w_{t_1}, w_{t_2}, ..., w_{t_k})$ and minibatches through simple first-order approximation. It is the total reduction in loss on a fixed validation example $I_{val}$ in the training process. We calculate the influence score with all the images in the train dataset $I_{train}$ and all the images in the validation dataset $I_{val}$. In equation 2, CLIP score computes correlation between CLIP image embedding of training images ($E_{I^{train}}$) and CLIP text embedding ($E_{T_c^{val}}$) for class $c$. Using IFT, we calculate the average of the scores of textual descriptions of each image class and then select the ten textual descriptions with the highest scores. We name these as *proponent texts*. Through these proponent texts, we can explain and understand each image class well.

## 3.3 Cross-Modal Transfer Classification

Cross-modal transferability (Zhang et al.) states that text inputs can work as good proxies to image inputs trained on a shared image-text embedding space obtained through contrastive learning. Based on this phenomenon, we can use texts as inputs instead of images for vision models trained with images if images and texts are in a shared embedding space.

We use CLIP (Radford et al., 2021) encoders to put the image and text in the same embedding space, which makes CLIP embeddings of both images and proponent texts the same size. After that, we feed these CLIP embeddings to a linear layer that has input dimension matched to the CLIP embedding size and output dimension matched to the number of classes to proceed with the classification task. When we train only with images, we feed the CLIP embeddings of images into the linear layer, calculate the loss through cross-entropy loss with the predicted label and ground truth label, and update the linear layer. After training with images, we do cross-modal transfer training with the CLIP embeddings of proponent texts. We calculate loss by giving weights according to the IFT of each proponent text. The weights are given by the IFT of each textual description divided by the sum of all IFTs. That is, the higher the IFT, the higher the weight, and the greater the loss for that text description. Detailed training algorithm is in Algorithm 1 in Appendix B. Our method has a low computational cost because we freeze CLIP encoder and only update the linear layer. For all datasets, it takes about 2 hours with NVIDIA 3090 GPU when training with only images and less than 30 minutes when we do cross-modal transfer training with proponent texts.

When we train the vision model additionally with proponent texts, the performance is improved compared to when we train with only images. It indicates that the proponent texts we set are textual descriptions that can explain each image class well. We can go further to see which texts the model focus on. In other words, it may provide an inherent explanation of the decision-making process of the black-box model.

## 4 Experiment

### 4.1 Experimental Setup and Details

**Datasets** We use nine image datasets for our experiments: CUB 200 2001 (Wah et al., 2011), Mini-imagenet (Vinyals et al., 2016)), CIFAR-10 (Krizhevsky et al., 2009), CIFAR-100 (Krizhevsky et al., 2009), OxfordPets (Parkhi et al., 2012), EuroSAT (Helber et al., 2019), Food101 (Bossard et al., 2014), 102flowers (Nilsback & Zisserman, 2008), and Describable Textures Dataset (DTD) (Cimpoi et al., 2014). For dataset partitions, we follow the official ones if provided, and when the official validation set is not provided, we use 20% of the training set as the validation set. If there is no officially divided train set and test set, we randomly divided the dataset to train, validation and test set. We report the dataset details including dataset paritions we use in Appendix I.

**Implementation Details** We use ViT/32 CLIP (Radford et al., 2021) encoder to put the image and text in the same embedding space. We train the linear model using stochastic gradient descent (SGD) with a mini-batch size of 64 with learning rate 0.1. We additionally train the last checkpoint of the vision model trained only with images with proponent texts.. We train total 30 epochs in all settings. Also, further implementation details are in Appendix H.

**Baseline** We compare our method with another method that uses large language model to extract text descriptions for image classes (Menon & Vondrick, 2023). This paper queries GPT-3 to extract text descriptors for each image class and provide zero-shot classification using text descriptors not just using raw class name. To the best of our knowledge, Menon & Vondrick (2023) is the only method that is closely related to our approach. Since baseline method (Menon & Vondrick, 2023) does not provide dataset partition information for the dataset they use, we reproduce the results by using code provided by the authors for all datasets, including datasets not used in Menon & Vondrick (2023). In addition, we include the performance reported in Menon & Vondrick (2023) as reference when we report the results.

| Dataset | CLIP Zero-Shot | Baseline(Reported) Menon & Vondrick (2023) | Baseline Menon & Vondrick (2023) | Baseline(GPT-3.5) Menon & Vondrick (2023) | Ours Zero-Shot | Only Images | Cross-Modal Transfer Classification |
|---|---|---|---|---|---|---|---|
| CUB 200 2011 | 38.540% | 52.57% | 51.881% | 52.969% | 53.227% | 71.332% | **74.525%** |
| OxfordPets | 81.132% | 83.46% | 84.636% | 85.580% | 88.679% | 91.644% | **93.396%** |
| CIFAR-10 | 88.800% | - | 88.610% | 89.320% | 89.470% | 94.320% | **94.820%** |
| CIFAR-100 | 61.680% | - | 64.010% | 63.999% | 64.570% | 77.830% | **78.250%** |
| EuroSAT | 30.815% | 48.94% | 41.444% | 32.630% | 39.148% | 94.926% | **96.037%** |
| Food101 | 80.620% | 83.63% | 83.419% | 83.644% | 83.452% | 85.848% | **86.950%** |
| Miniimagenet | 81.630% | - | 84.280% | 84.780% | 85.320% | 91.980% | **92.480%** |
| 102flowers | 58.730% | - | 67.643% | 66.670% | 69.109% | 96.459% | **97.192%** |
| DTD | 43.085% | 44.26% | 43.245% | 37.660% | 48.989% | 72.074% | **74.043%** |

Table 1: Accuracies for test images when training using only images (Only Images) and cross-modal transfer training with proponent texts (Cross-Modal Transfer Classification), along with zero-shot setting (Ours Zero-shot). Additionally, we report the reproduced results with authors' code (Baseline) and results when the baseline method use GPT-3.5, LLM the same as our method (Baseline(GPT-3.5)). We also include the performance reported Menon & Vondrick (2023) (Baseline(Reported)). Furthermore, we also report the performance of the CLIP in zero-shot setting (CLIP zero-shot). The best performing ones in bold font and underline represents the best performance in zero-shot setting.

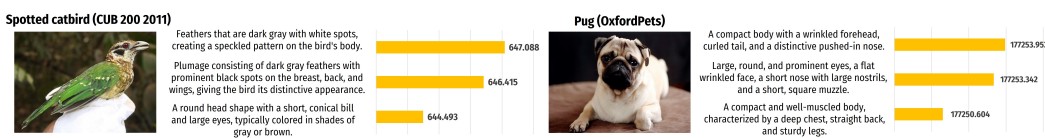

Figure 2: Examples of proponent images, proponent texts and IFT for two classes.

## 4.2 RESULT

Table 1 reports the performance of cross-modal transfer classification and zero-shot setting to evaluate our text descriptions to compare with the baseline method. In addition, since the baseline method uses GPT-3, for fair comparison, we report the performance when the baseline method uses GPT-3.5 by using provided code from the authors.

Table 1 shows that our method outperforms the baseline in zero-shot setting on most datasets. Even when the baseline method uses the same large language model, GPT-3.5, our method performs better. For Food101 dataset, the baseline performance using GPT-3.5 is better than ours, but the difference is very small. For EuroSAT dataset, our performance is worse than baseline and we conjecture that the EuroSAT dataset consists of satellite images and the size of the image is small, so even if more detailed explanations are provided, these are not helpful for the model when classifying the images. Especially, for DTD dataset, our method's accuracy in zero-shot setting is higher by approximately 5% compared to baseline. For OxfordPets dataset, our zero-shot performance is approximately 4% higher than baseline. The difference between the reproduced performance and reported performance of baseline method for EuroSAT dataset may be due to the differences in class labels. For example, if the original class name is 'residential', baseline method changes class name to 'residential buildings or homes or apartments' when we check it from authors' code. This result demonstrates that our textual descriptions can explain each image class better than the baseline method. Also, we can identify that our 2-stage prompting with Wikipeida url is effective as our method performs better even when the baseline method uses the same large language model as ours. To investigate why our method performs better than baseline, we visualize the embedding of each textual description and the embeddings of the same class images with t-SNE (Van der Maaten & Hinton, 2008). As one can see in Figure 3, the embeddings of our textual descriptions are much closer to the embeddings of the images than those of the baseline method. More visualization results are in E.

Furthermore, for all datasets, the performance is better when the vision model is cross-modal transfer trained with the proponent texts than when only trained with images. In particular, for CUB 200 2011 dataset, performance increases by approximately 3% when cross-modal transfer trained with proponent texts compared to when trained with images alone. For other datasets, performances increase by 1-2% compared to when trained only with images. This result also supports that our proponent texts are texts that can explain each image class well and the prediction of the black-box model by using IFT as training loss weights. Figure 2 provides examples of proponent images and proponent texts, and more examples are in Appendix C.

We hypothesize that the reason why cross-modal transfer training with proponent texts performs better than when we train only with images may be because when training the model using images

| Dataset | Only Images | Cross-Modal Transfer Classification(No Wiki) | Cross-Modal Transfer Classificatio(IF) | Cross-Modal Transfer Classification(CLIP) | Cross-Modal Transfer Classification(IFT & Wiki) |
|---|---|---|---|---|---|
| CUB 200 2011 | 71.332% | 72.092% | 72.834% | 72.696% | **74.525%** |
| OxfordPets | 91.664% | 92.453% | 92.722% | 92.318% | **93.396%** |
| CIFAR-10 | 94.320% | 94.490% | 94.680% | 94.730% | **94.820%** |
| CIFAR-100 | 77.830% | 77.410% | 77.910% | 77.970% | **78.250%** |
| EuroSAT | 94.296% | 94.333% | 94.222% | 94.407% | **96.037%** |
| Food101 | 85.848% | 86.832% | 85.974% | 85.934% | **86.950%** |
| Miniimagenet | 91.980% | 91.570% | 92.040% | 92.020% | **92.480%** |
| 102flowers | 96.459% | 96.703% | 96.460% | 96.337% | **97.192%** |
| DTD | 72.074% | 73.830% | 73.989% | 73.830% | **74.043%** |

Table 2: Ablation study accuracies on Cross-Modal Transfer Classification. The proponent text is determined using only influence score(Cross-Modal Transfer Classification(IF)) or CLIP score (Cross-Modal Transfer Classification(CLIP)) instead of IFT (Cross-Modal Transfer Classification(IFT & Wiki)). We also report the performance when we do not provide Wikipedua url to GPT-3.5 (Cross-Modal Transfer Classification(No Wiki)).

and proponent texts, the model utilize and learn both of the features of images and contexts of proponent texts. Since the text context is information containing external knowledge extracted through large language models, the model may learn more relevant information about image classes.

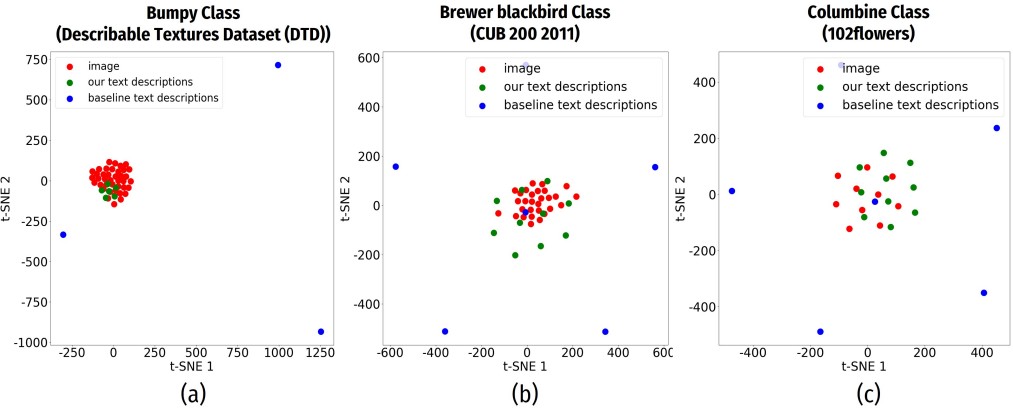

Figure 3: Visualization of the embeddings of textual descriptions of our method and baseline method and the embeddings of the same class images with t-SNE. (a) Bumpy class of DTD dataset (b) Brewer blackbird class of CUB 200 2011 dataset (c) Columbine class of 102flowers dataset

### 4.3 ABLATION STUDIES

First, to verify our IFT is appropriate as a score metric for texts, we compare the performance of cross-modal transfer training the vision model with the selected proponent texts when either the influence score or CLIP score is used instead of the IFT and the result is in Table 2. For all datasets, compared to when cross-modal transfer training with proponent texts selected using IFT, the performance is lower when only the influence score or CLIP score is used.

We also show examples of proponent texts that changes when using only the influence score or CLIP score, and IFT in Figure 4. In this figure, one can see that proponent texts selected using only the CLIP score are just error message, and rather about the decoration of the bus' appearance and the material of the bus' wheel. It is hard to say that this information is helpful visual cues to classify the bus images. The proponent texts selected using the influence score contains information about the appearance of not only the bus, but also about the structure inside the bus. Since the influence score is fixed after once calculated, the score is the same for all texts. So, when cross-modal transfer training with proponent texts selected using the influence score, all of them are calculated with the same loss weight. On the other hand, all of the proponent texts determined using IFT provides clues to classify the bus images, and one can see what factors help the model to make a decision.

Through IFT, we can resolve issues mentioned in baseline methods, such as visual cues that do not help for models when classifying images, or the same text being repeated. It can be seen that IFT is appropriate as a score metric for texts. Further analysis is in Appendix F to demonstrate that IFT does a role to select helpful visual cues to explain each image class among several texts.

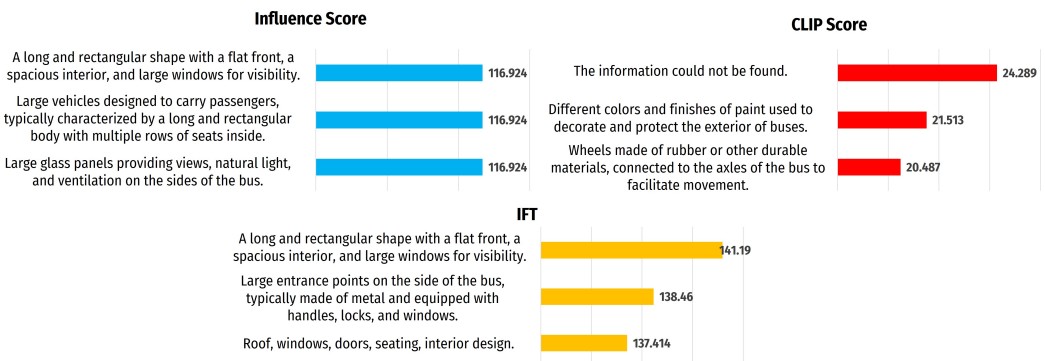

Figure 4: Examples of scores and the selected proponent texts when using influence score(Influence Score), CLIP score(CLIP Score), and IFT. (Bus class of CIFAR-100 dataset)

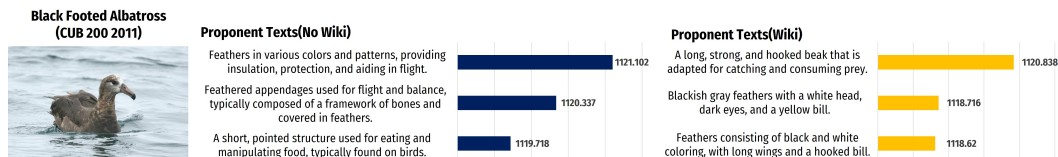

Figure 5: Examples and scores of the selected proponent texts when we provide Wikipedia url to GPT-3.5 or not. (Black footed albatross class of CUB 200 2011 dataset)

Second, to demonstrate our 2-stage prompting with Wikipedia url is effective, we compare the performance of cross-modal transfer training the vision model with the selected proponent texts when we provide Wikipedia url to GPT-3.5 or not and the result is in Table 2. For all datasets, the performance of cross-modal transfer classification with selected proponent texts when Wikipedia url is provided to GPT-3.5 is better than when Wikipedia url is not provided.

We provide examples of proponent texts when Wikipedia url is provided or not to illustrate the cause of this. In Figure 5, the selected proponent texts when we does not provide Wikipedia url give general information related to bird rather than specific information of the corresponding class: Feathered appendages used for flight and balance, Feathers in various colors and patterns, providing insulation, protection, and aiding in flight. However, when we provide Wikipedia url, we can see textual descriptions that describe features related to that class well: Blackish gray feathers with a white head, dark eyes, and a yellow bill; Feathers consisting of black and white coloring, with long wings and a hooked bill.

These show that our 2-stage prompting with Wikipedia url for large language models is effective.

## 4.4 HUMAN EVALUATION

|  | Menon & Vondrick (2023) | Ours | Tie |
|---|---|---|---|
| Relevant | 0.144% | **0.485%** | 0.371% |
| Informative | 0.060% | **0.790%** | 0.150% |
| Helpful | 0.066% | **0.670%** | 0.263% |

Table 3: Human evaluation result to qualitatively evaluate our textual descriptions compared with Menon & Vondrick (2023). Our text descriptions are significantly preferred in all factors.

We conduct human evaluation to qualitatively evaluate our textual descriptions compared with Menon & Vondrick (2023). We show one image and two corresponding text descriptions from Menon & Vondrick (2023) and ours for each question and ask to choose which of the two text descriptions is more relevant to the image (Relevant row in Table 3); more informative to describe the image (Informative row in Table 3); more helpful to identify the image (Helpful row in Table 3). We asked five people about a total of 100 randomly chosen classes from all nine datasets. As one can see in Table 3, our text descriptions were significantly preferred in all factors.

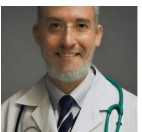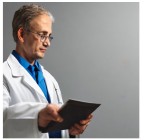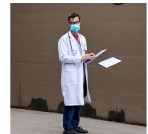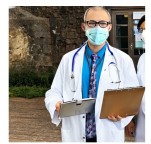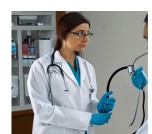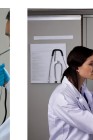

**Raw Descriptions**

A photo of doctor

**Baseline Descriptions**

doctor, who wearing a white lab coat,

whose has stethoscope around the neck,

who wearing a face mask,

...

**Ours Descriptions**

doctor, who put on white coat, scrubs, stethoscope, ...

who has tool used by doctors to listen to internal body ...

who has stethoscope, syringe, scalpel, forceps, ...

...

(a)                               (b)                               (c)

Figure 6: Generated images of doctor using Stable Diffusion v2. (a) Generated images when using prompts used in CLIP. (b) Generated images when using text descriptions of baseline method. (b) Generated images when using text descriptions of our method.

We assume the reason why our textual descriptions are preferred in all elements compared to those of baseline may be because our textual descriptions have more detailed explanations. For blue jay bird for CUB 200 2011 dataset, descriptions for baseline method are: a black beak; a blue bird. However, our textual descriptions provide more detailed descriptions, such as not just black beak, but what shape of beak (short, sturdy, and slightly curved beak ), not just blue bird, but bright blue feather, and blue-tinged tail. The examples of human evaluation questions are in Appendix D.

### 4.5 Solving Model Bias

Bias is a widespread problem in various fields such as word embedding, text-to-image generation model, and machine learning models (Naik & Nushi, 2023; Bolukbasi et al., 2016). One of such biases is a gender bias related to occupation. For example, there is a stereotype that a CEO or computer programmer is a man and a housewife is a woman.

It has been found that one of popular text-to-image generation model, Stable Diffusion (Rombach et al., 2022) has a gender bias for the occupation 'doctor'. As one can see in Figure 6 (a), when we generate images using prompts used in CLIP (Radford et al., 2021), a photo of doctor, Stable Diffusion model only generated images of male doctors. With our 2-stage prompting with Wikipedia url, we can get descriptions about the doctor. With these descriptions, we make sentence used to generate the images using the following form with generated descriptions: doctor, who has/is/wearing/put on {descriptions}. In Figure 6 (c), we can see that images generated with our text descriptions include not only male but also female doctor. For comparison, we also conducted text-to-image generation experiments with the descriptions of the doctor generated by the baseline method. As one can see in Figure 6 (b), images of doctors are generated, but we cannot see images of female doctors.

Through this, we show that using text descriptions generated by our method can solve bias in text-to-image generation model quite effectively without modifying prompts directly by humans. It is also observed that our method can generate images with reduced bias even when compared to CLIP and baseline method. We hypothesize the reason why images with reduced gender bias are generated with our textual descriptions compared to baseline may be that our descriptions are more detailed and relevant, so the text-to-image generation model interprets these descriptions more neutral to the gender. More examples of solving gender bias related to occupations such as computer programmer and CEO are in Appendix G.

## 5 Conclusion

We propose a novel approach that can understand the data by extracting textual descriptions using 2-stage prompts with Wikipedia url to utilize external knowledge bases through large language models. Using the proposed IFT defined as the sum of the influence score and the CLIP score, we can determine *proponent text*, which is informative text for explaining each image class. Furthermore, We propose a novel benchmark task named cross-modal transfer classification. By cross-modal transfer training the vision model with proponent texts, we can improve the performance even more than when using only images. We also show solving bias. Through our approach, we can understand the data in terms of language and may provide inherent interpretability of model prediction.

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

## A   2-STAGE PROMPTING WITH WIKIPEDIA URL

Since large language models generate different quality of answers depending on prompting, we provide more detailed implementation details of our 2-stage prompting with the wikipedia url. We refer to the setting of baseline method (Menon & Vondrick, 2023). We used the prompting structure of Q: A: and provided GPT-3.5 desired output examples.

First, when we are extracting components of each class, we remind the form of questions:

> Q : Can you tell me the components of {class name} from the perspective of appearance?
>
> A :

Prior to the above question, we provide additional examples like this:

Q : Can you tell me the components of american bulldog from the perspective of appearance?

A :  1. Coat Type and Texture

2. Coat Color

3. Body Build

4. Size

5. Head

6. Muzzle and Nose

7. Eyes

8. Ears

9. Tail

10. Legs and Paws

11. Coat Patterns

12. Facial Features

13. Unique Breed Traits

Since we provided these examples to GPT-3.5, when we ask a question, the answers in the order of 1, 2, 3 are given. So we can the desired answers simply remove the numbers.

We also provide detailed process of extracting the components of each class with examples in Figure 7.

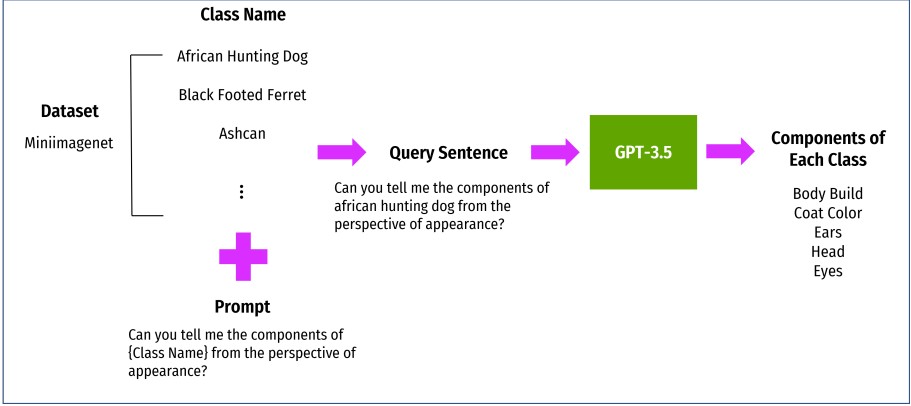

Figure 7: First step to extract textual descriptions. Question prompt is the form of query for GPT-3.5 which requires class name. From the query sentence, we can get the component of each class.

Second, when we are extracting textual descriptions of each class using the components of each class and the wikipedia url, we remind the form of questions:

Q : Please summarize the information of appearance about {components of each class} in this {wikipedia url} in one line composed of nouns.
If you couldn't find related information, you must answer general information you know.
A :

Prior to the above question, we also provide additional examples like this:

Q : Please summarize the information of appearance about nose in this url https://en.wikipedia.org/wiki/American_Bulldog in one line composed of nouns. If you couldn't find related information, you must answer general information you know like the above questions.
A : A short to medium-length muzzle with a nose that can be black, brown, or pigmented, often matching the coat color, and it is a distinctive feature on the breed's square-shaped head.

Since we asked for a "composed of nouns" answer, we can get a detailed and informative answer without refining.

We also provide detailed process of extracting textual descriptions of each class using the components of each class with examples in Figure 8.

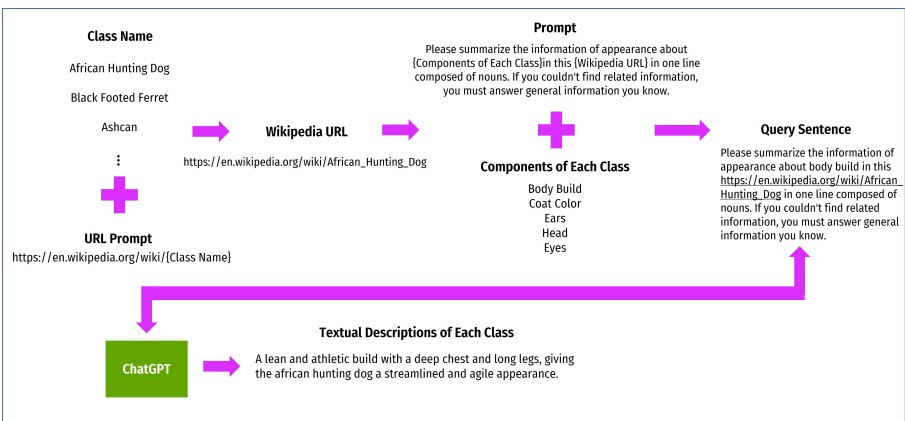

Figure 8: Second step to extract textual descriptions. In this step, question query requires the components of each class and Wikipedia url. The Wikipedia url here is used as a tool to utilize external knowledge bases to fill the lack of detailed knowledge with LLMs. As a result, we get the textual descriptions of each class.

## B TRAINING ALGORITHM

In this section, we provide detailed training algorithm of cross-modal transfer classification.

---

**Algorithm 1** Training Algorithm of Cross-Modal Transfer Classification

---

**Require:** $(I_{c_i}, L_{c_i})$ : pairs of image and label for class $c_i$
**Require:** $(T_{c_i}, L_{c_i}, IFT_{c_i})$ : pairs of proponent text, label and IFT for class $c_i$
**Require:** $CLIP$ : CLIP model
**Require:** $LinearLayer(d_{CLIP}, |c|)$ : Linear layer that matches the CLIP embedding size and the number of classes
**Require:** $CEL$ : Cross-entropy loss function
1: $E_I \leftarrow []$ ▷ List to store image embeddings
2: $E_T \leftarrow []$ ▷ List to store text embeddings
3: $IFT \leftarrow []$ ▷ List to store IFT
4: **for** each class $c_i$ **do**
5:     **for** each pair $(I_{c_i}, L_{c_i})$ of image and label **do**
6:         $E_I \leftarrow E_I + [CLIP(I_{c_i})]$
7:     **end for**
8:     **for** each pair $(T_{c_i}, L_{c_i}, IFT_{c_i})$ of text, label and IFT **do**
9:         $E_T \leftarrow E_T + [CLIP(T_{c_i})]$
10:         $IFT \leftarrow IFT + [IFT_{c_i}]$
11:     **end for**
12: **end for**
13: Initialize linear layer and freeze CLIP
14: **for** each epoch **do**
15:     **for** each batch **do**
16:         Extract image batch embeddings and labels
17:         $L_c^{(\text{pred})} \leftarrow \text{LinearLayer}(E_I; W, b)$
18:         $loss = CEL(L_c^{(\text{pred})}, L_{c_i})$
19:     **end for**
20: **end for**
21: **for** each epoch **do**
22:     **for** each batch **do**
23:         Extract text batch embeddings, IFTs and labels
24:         Calculate text weights $W_{T_{c_i}}$ for each proponent texts for each class using $IFT$ (e.g., $W_{T_{c_i}} = \frac{IFT_{c_i}}{\sum IFT_c}$)
25:         $L_c^{(\text{pred})} \leftarrow \text{LinearLayer}(E_T; W, b)$
26:         $loss = W_{T_{c_i}} * CEL(L_c^{(\text{pred})}, L_{c_i})$
27:     **end for**
28: **end for**
29: **return** Trained linear layer

---

## C PROPONENT TEXTS AND IFT

In this section, we provide additional examples of proponent texts and IFT.

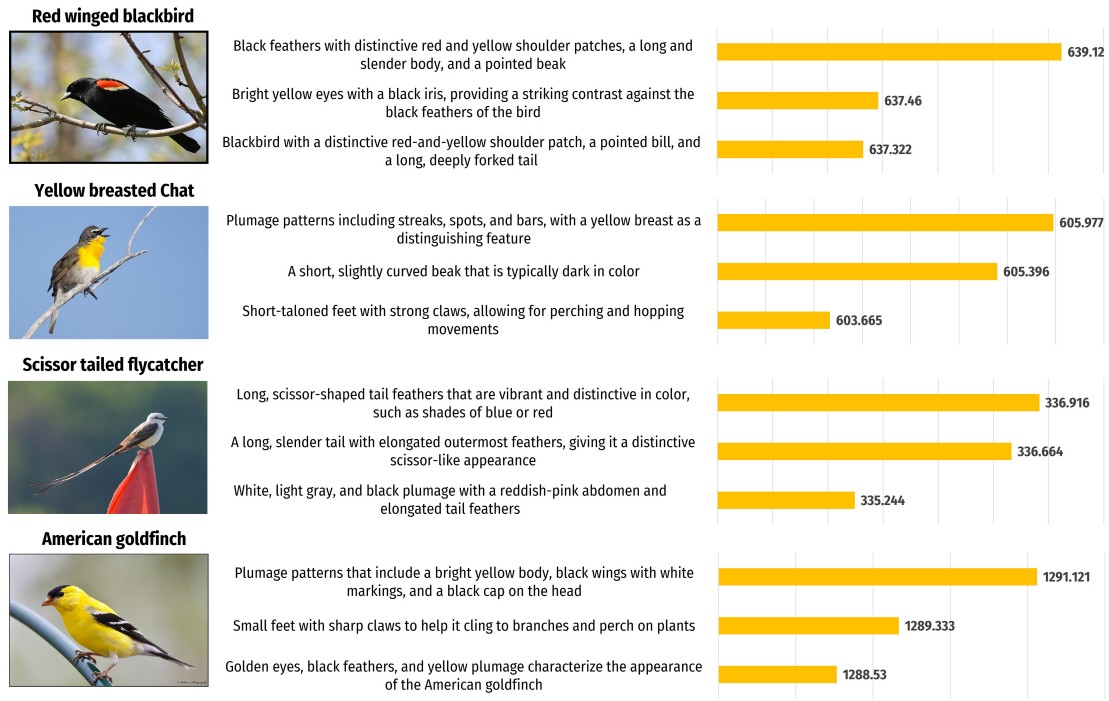

Figure 9: Examples of proponent images, proponent texts and IFT for four classes of CUB 200 2011 dataset.

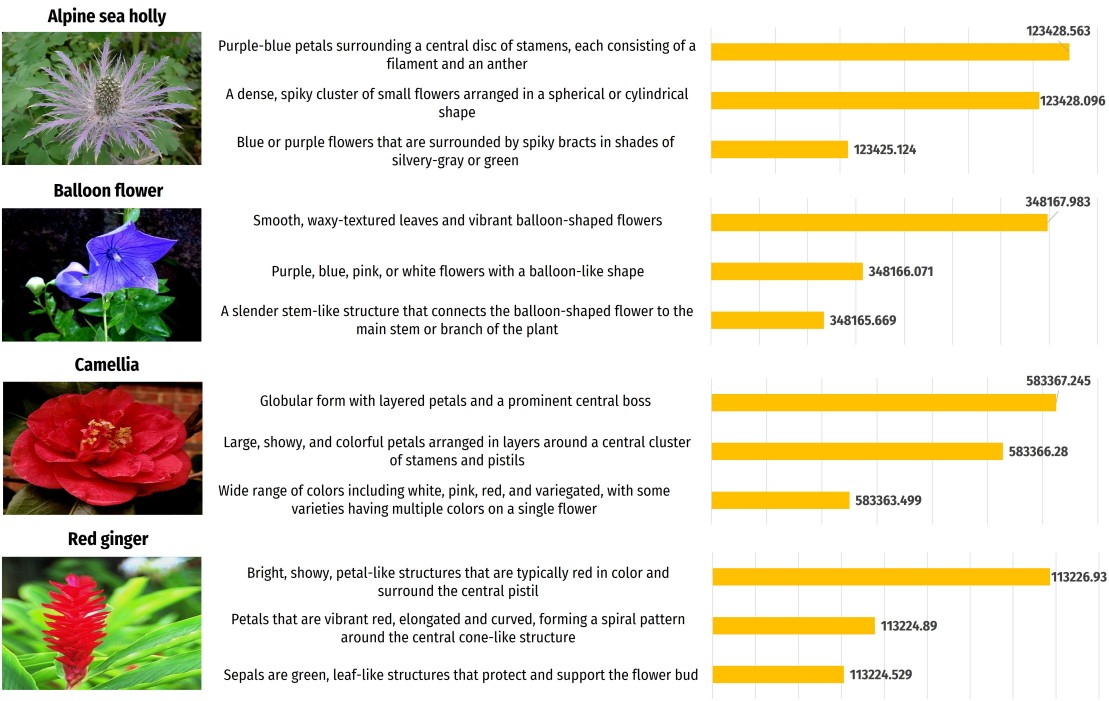

Figure 10: Examples of proponent images, proponent texts and IFT for four classes of 102flowers dataset.

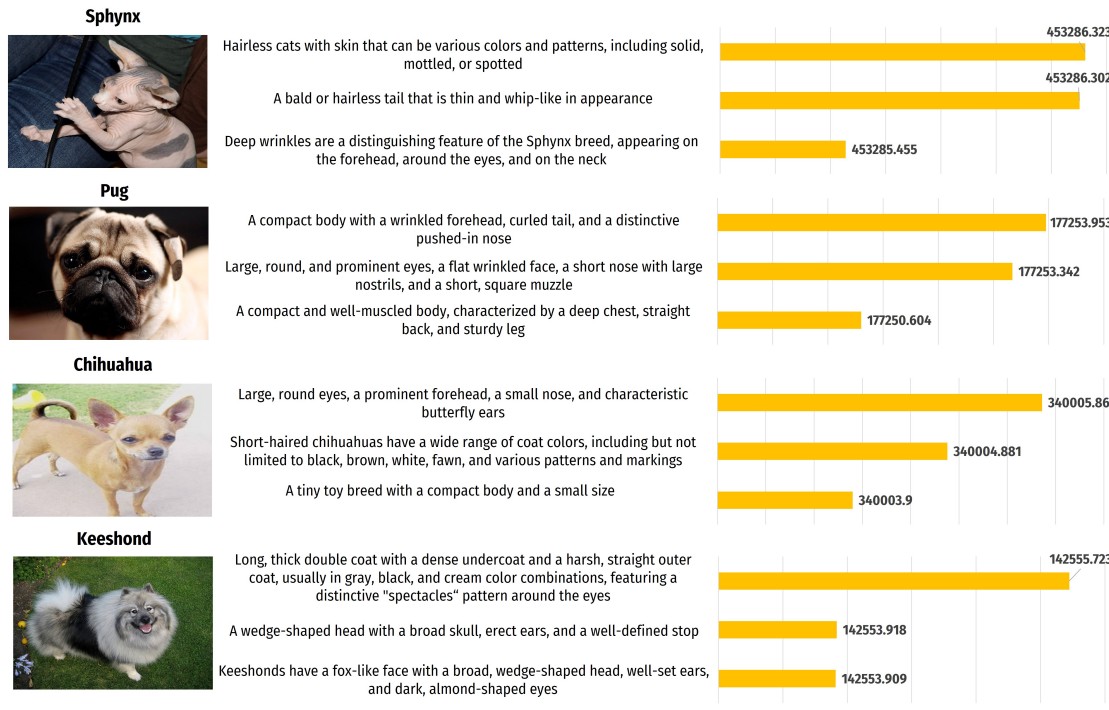

Figure 11: Examples of proponent images, proponent texts and IFT for four classes of OxfordPets dataset.

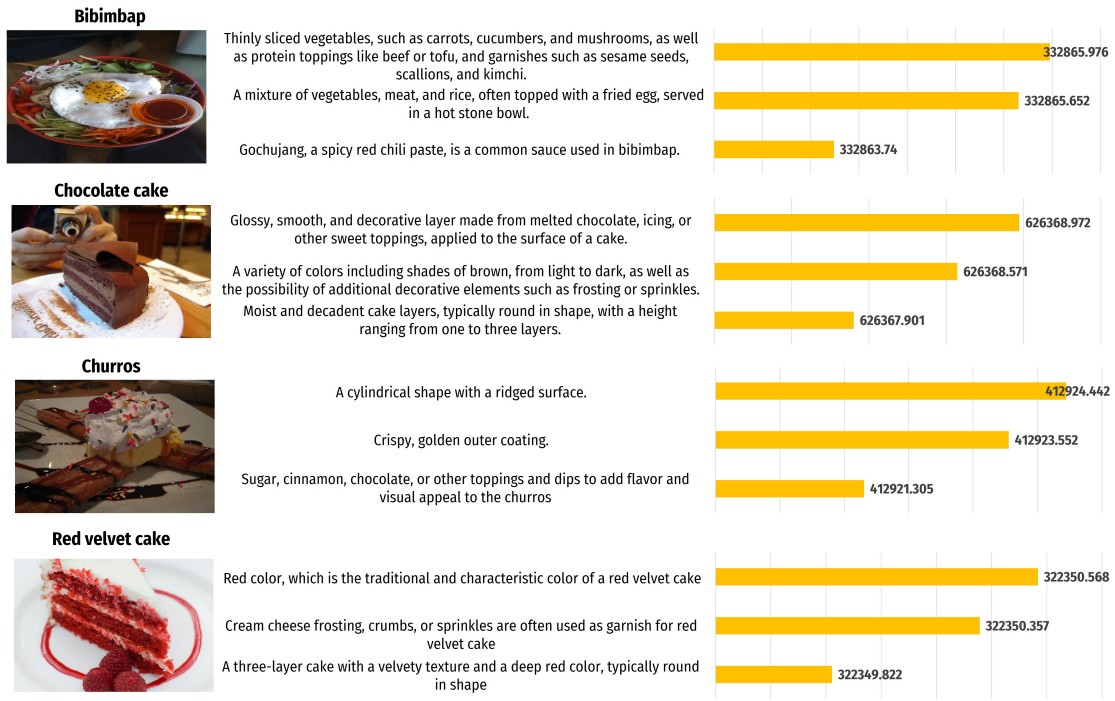

Figure 12: Examples of proponent images, proponent texts and IFT for four classes of Food101 dataset.

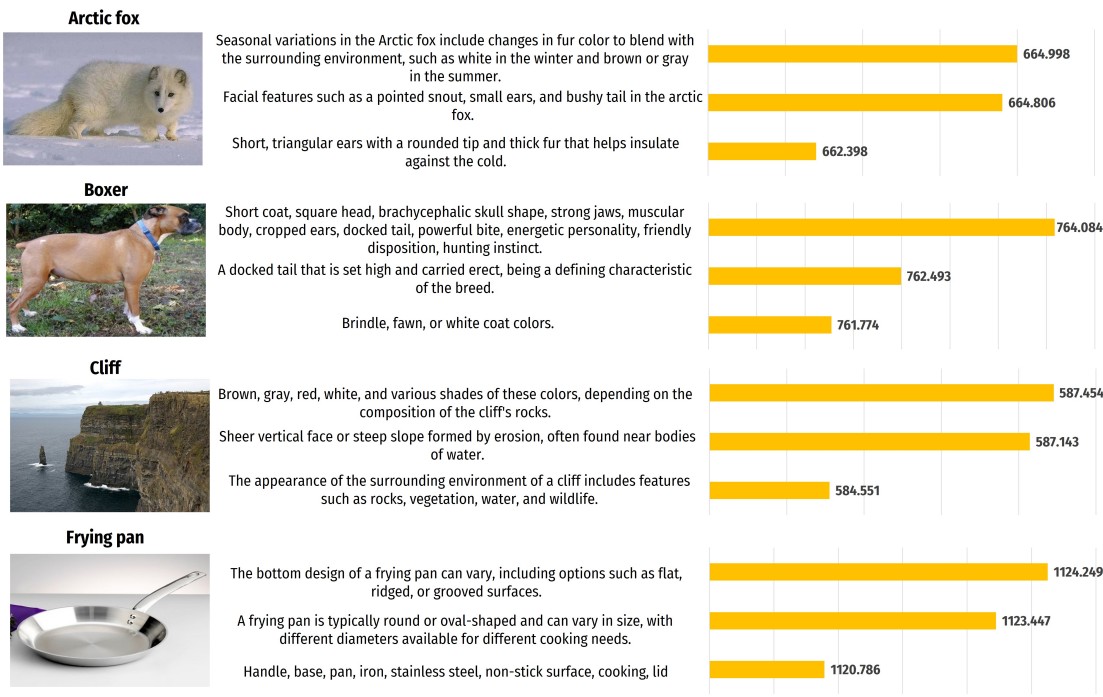

Figure 13: Examples of proponent images, proponent texts and IFT for several classes of Miniimagenet dataset.

# D  HUMAN EVALUATION EXAMPLES

In this section, we provide some qualitative examples of human evaluation questions that comparing our textual descriptions with those of baseline.

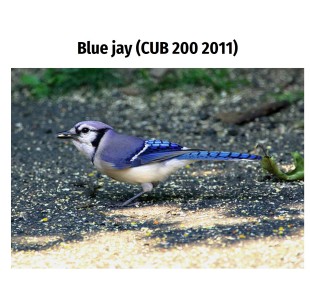

**Baseline**

- a blue bird
- a white chest
- a black neck and head
- a black beak
- a long tail
- black legs

**Ours**

- Crests consisting of a prominent crown of feathers on the top of their heads, often bright blue in color
- Facial features including a short, robust beak, a crest on the head, and vibrant blue feathers
- A short, sturdy, and slightly curved beak
- A long, blue-tinged tail with white patches on the underside, serving as a visual and behavioral signal in communication
- White, black, and blue plumage with distinctive crests, markings, and patterns, including a white underbelly, black collar, and blue wings and tail feathers.
- Long legs and strong feet, with sharp claws for perching and grasping.

Figure 14:  Blue jay class of CUB 200 2011 dataset

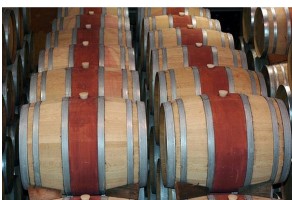

**Barrel (Miniimagenet)**

**Baseline**

- wooden or metal construction
- metal bands around the circumference
- a lid or bung hole
- a spigot or tap for draining liquid
- a handle for carrying or moving the barrel

**Ours**

- Curved and cylindrical form that is wider in the middle and tapers towards the ends
- The color of barrels can vary depending on the material they are made of, such as wood, metal, or plastic
- Metal hoops used to hold the staves of a barrel together, providing structural support
- Materials like cork, wood, or metal used to seal and secure the end of a barrel
- Multiple wooden planks or strips used to create the curved shape of a barrel and held together with metal or wooden hoops
- Oak barrel label with information about the contents, producer, and aging process

Figure 15: Barrel class of Miniimagenet dataset

**Lined
(Describable Textures Dataset (DTD))**

**Baseline**

- a series of parallel lines
- can be straight or curved
- may be of different colors
- may be of different widths
- may be of different thicknesses

**Ours**

- Stripes, patterns, or markings that create a visual effect of lines
- Parallel lines that create texture or pattern in a surface
- Stripes, patterns, and designs that can be consistent or inconsistent in terms of width, spacing, and alignment
- Detailed, intricate, and fine-lined patterns that are often used for artistic and decorative purposes on various surfaces

Figure 16: Lined class of DTD dataset

## E   VISUALIZATION RESULTS WITH T-SNE

In this section, we provide additional visualization results of our text descriptions and those of the baseline with the same class images.

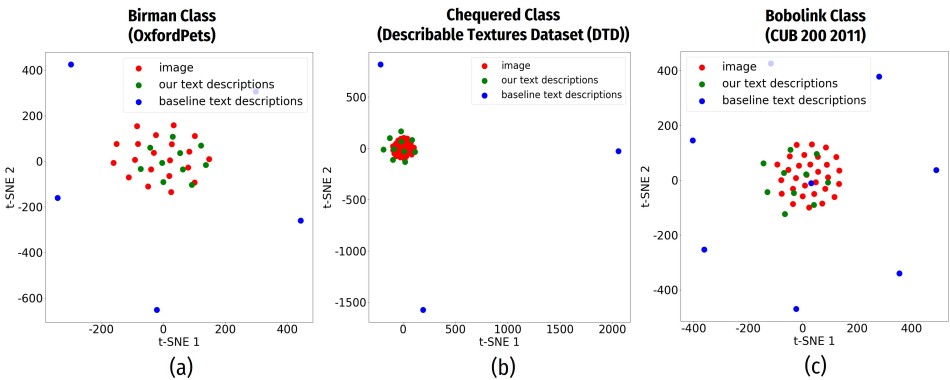

Figure 17: Visualization of the embeddings of textual descriptions of our method and baseline method and the embeddings of the same class images with t-SNE. (a) Birman class of Oxford-Pets dataset (b) Chequered class of DTD dataset (c) Bobolink class of CUB 200 2011 dataset

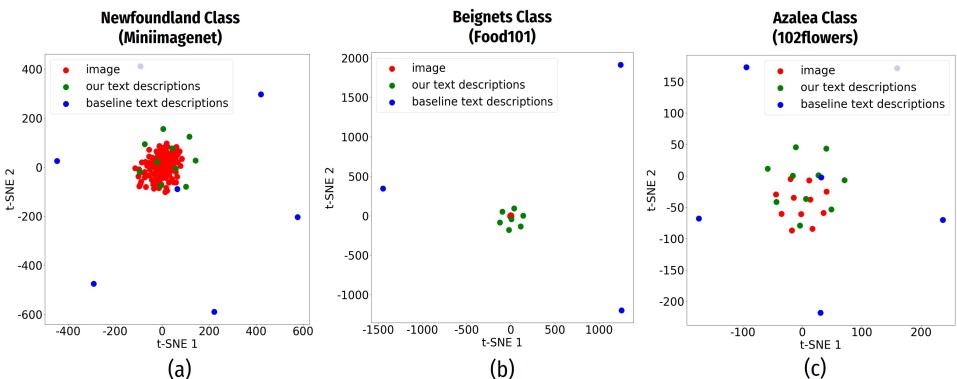

Figure 18: Visualization of the embeddings of textual descriptions of our method and baseline method and the embeddings of the same class images with t-SNE. (a) Newfoundland class of Miniimagenet dataset (b) Beignets class of Food101 dataset (b) Azalea class of 102flowers dataset

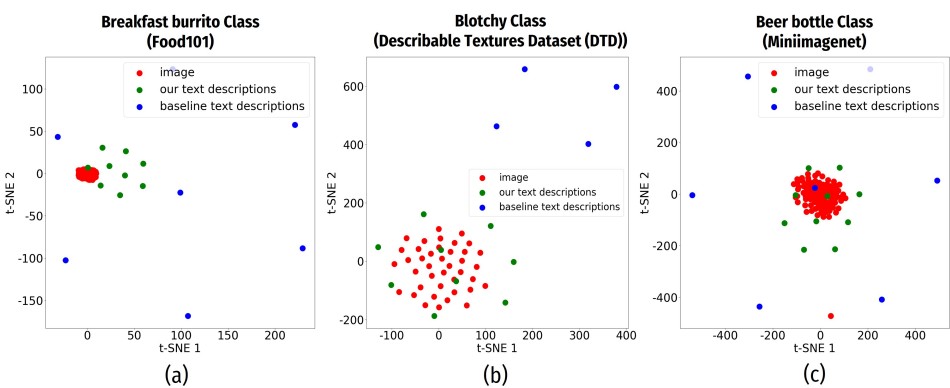

Figure 19: Visualization of the embeddings of textual descriptions of our method and baseline method and the embeddings of the same class images with t-SNE. (a) Breakfast burrito class of Food101 dataset (b) Blotchy class of DTD dataset (b) Beer bottle class of Miniimagenet dataset

## F ANALYSIS

In this section, to demonstrate that IFT can select texts that can describe each image class well, we show textual descriptions selected as proponent text and text that is not.

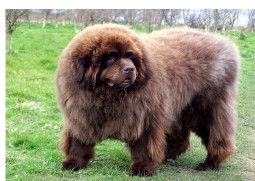

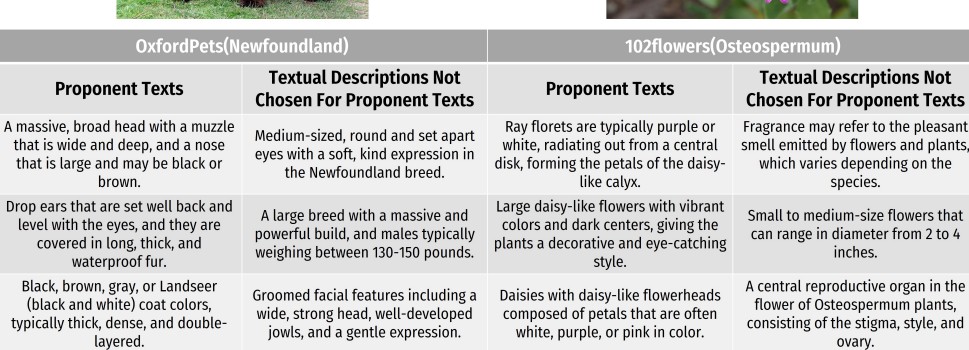

| OxfordPets(Newfoundland) | | 102flowers(Osteospermum) | |
|---|---|---|---|
| **Proponent Texts** | **Textual Descriptions Not Chosen For Proponent Texts** | **Proponent Texts** | **Textual Descriptions Not Chosen For Proponent Texts** |
| A massive, broad head with a muzzle that is wide and deep, and a nose that is large and may be black or brown. | Medium-sized, round and set apart eyes with a soft, kind expression in the Newfoundland breed. | Ray florets are typically purple or white, radiating out from a central disk, forming the petals of the daisy-like calyx. | Fragrance may refer to the pleasant smell emitted by flowers and plants, which varies depending on the species. |
| Drop ears that are set well back and level with the eyes, and they are covered in long, thick, and waterproof fur. | A large breed with a massive and powerful build, and males typically weighing between 130-150 pounds. | Large daisy-like flowers with vibrant colors and dark centers, giving the plants a decorative and eye-catching style. | Small to medium-size flowers that can range in diameter from 2 to 4 inches. |
| Black, brown, gray, or Landseer (black and white) coat colors, typically thick, dense, and double-layered. | Groomed facial features including a wide, strong head, well-developed jowls, and a gentle expression. | Daisies with daisy-like flowerheads composed of petals that are often white, purple, or pink in color. | A central reproductive organ in the flower of Osteospermum plants, consisting of the stigma, style, and ovary. |

Figure 20: Textual descriptions determined as proponent texts and textual description that are not for newfoundland class of OxfordPets dataset and osteospermum class of 102flowers dataset.

Figure 20 shows texts not selected as proponent texts among the textual descriptions of the newfoundland class of OxfordPets dataset and osteospermum class of 102flowers dataset. In the case of newfoundland class of OxfordPets dataset, textual descriptions that are not selected as proponent texts do not help classify the newfoundland images, one of the types of dogs, but contain general information such as expression or weight of the newfoundland dog. In the case of osteospermum class, one of the types of flowers, of 102 flowers dataset, textual descriptions that are not selected as proponent texts contain information about fragrance, how much it grows and reproductive organ. This information does not provide the model helpful cues to classify the image. Through this, we see that the IFT plays a role in ensuring that the proponent texts only contain information that helps the model to classify the image well.

## G SOLVING MODEL BIAS

In this section, we provide more examples of solving gender bias related to occupation in text-to-image generation task through our method. There is a stereotype that a CEO or computer programmer is a man.

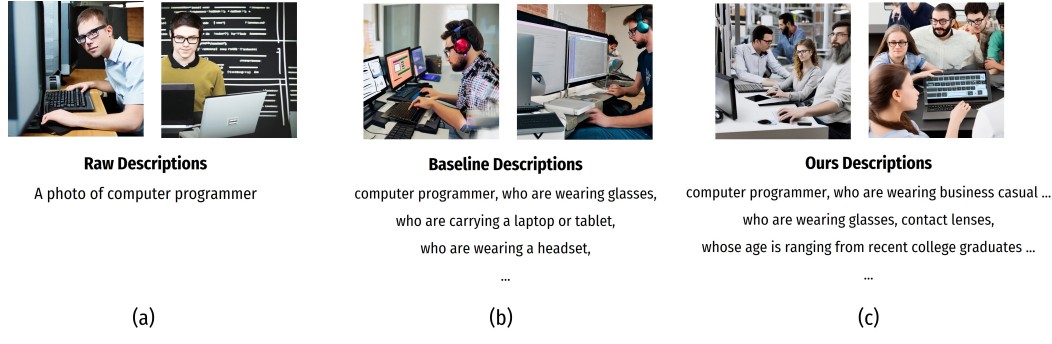

| Raw Descriptions | Baseline Descriptions | Ours Descriptions |
|---|---|---|
| A photo of computer programmer | computer programmer, who are wearing glasses, who are carrying a laptop or tablet, who are wearing a headset, … | computer programmer, who are wearing business casual … who are wearing glasses, contact lenses, whose age is ranging from recent college graduates … … |
| (a) | (b) | (c) |

Figure 21: Generated images of computer programmer using Stable Diffusion Stable Diffusion. (a) Generated images when using prompts used in CLIP. (b) Generated images when using text descriptions of baseline method. (b) Generated images when using text descriptions of our method.

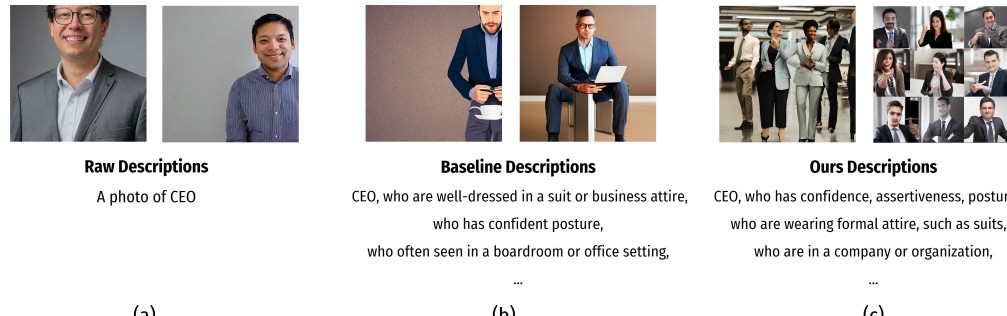

**Figure 22:** Generated images of CEO using Stable Diffusion Stable Diffusion. (a) Generated images when using prompts used in CLIP. (b) Generated images when using text descriptions of baseline method. (b) Generated images when using text descriptions of our method.

In Figure 21 (a), when we generate images using prompts used in CLIP (Radford et al., 2021), a photo of computer programmer, Stable Diffusion model only generated images of male computer programmers. In Figure 21 (b), with descriptions of the computer programmers generated by baseline method (Menon & Vondrick, 2023), still images of the male computer programmer are generated. In Figure (c), with our textual descriptions generated using 2-stage prompting with Wikipedia url, we can see that generated images include not only male but also female computer programmer. In Figure 22, in the case of CEO, it can also be seen that the images generated with our textual descriptions have reduced bias compared to CLIP or baseline methods.

## H    IMPLEMENTATION DETAILS

When we train the linear layer, we use CosineAnnealingLR learning scheduler with maximum number of iterations of 200. Additionally, we do not use any image augmentation strategies during training and inference.

To calculate influence score, we need checkpoints. So, we train the pretrained Resnet34 with stochastic gradient descent (SGD) and mini-batch size 64. The learning rate starts from 0.1 and is divided by 10 every 30 epoch and we terminate our training at 200th epoch. Since we can not use all checkpoints, we use checkpoints every 10 epoch.

## I    DATASET DETAILS

We use a total of nine datasets. Additionally, we report the size of the train set, validation and test set of the datasets we use in table 4.

**CUB 200 2001 (Wah et al., 2011)** This dataset contains a total of 200 bird species, with each species having around 30 images. It provides a challenging benchmark for image classification tasks due to the large number of species and the variability in lighting, pose, and background in the images.

**Miniimagenet (Vinyals et al., 2016)** This dataset consists of 100 classes, each containing 600 images. The classes are drawn from a larger dataset called ImageNet (Deng et al., 2009), which contains over a million images across thousands of classes. It was introduced in Vinyals et al. (2016) as a benchmark dataset for few-shot learning, but we use 500 images for each class as a train set and 100 images as a test set.

**CIFAR-10 (Krizhevsky et al., 2009)** This is a widely-used dataset in machine learning research, consisting of 60,000 32x32 color images in 10 classes. This dataset is split into a training set of 50,000 images and a test set of 10,000 images, with each image belonging to one of 10 classes.

**CIFAR-100 (Krizhevsky et al., 2009)** This is an extension of CIFAR-10 dataset, consisting of 60,000 32x32 color images in 100 classes. This dataset is also divided into a training set of 50,000 images and a test set of 10,000 images.

**OxfordPets (Parkhi et al., 2012)** The dataset is a collection of approximately 7,349 images containing 37 different breeds of cats and dogs. It is commonly used for tasks such as image classification and object detection in computer vision research. .

**EuroSAT (Helber et al., 2019)** This dataset is used for classifying geographical land cover types based on satellite imagery. It comprises 10 distinct geographical landscape classes, including categories like forests, cropland, roads, buildings, and rivers, among others. It has 27,000 satellite images.

**Food101 (Bossard et al., 2014)** The dataset is a widely used collection of food images that is primarily employed for food recognition and image classification tasks. It consists of approximately 101,000 images, each depicting a specific food item from one of the 101 distinct food categories. These categories cover a diverse range of foods, including various cuisines, dishes, and ingredients.

**102flowers (Nilsback & Zisserman, 2008)** The dataset is a collection of flower images, consisting of 102 different categories, each representing a distinct species or type of flower. The dataset contains a substantial number of flower images, typically around 40 to 258 images per category, resulting in a total of over 8,000 images.

**Describable Textures Dataset (DTD) (Cimpoi et al., 2014)** This dataset is a collection of diverse texture images, containing 47 different categories with distinct visual features and structures. It has a total of 5,640 high-resolution images.

| Dataset | Classes | Train Size | Validation Size | Test Size |
|---|---|---|---|---|
| CUB 200 2011 | 200 | 4800 | 1194 | 5794 |
| Miniimagenet | 100 | 40000 | 10000 | 10000 |
| CIFAR-10 | 10 | 40000 | 10000 | 10000 |
| CIFAR-100 | 100 | 40000 | 40000 | 10000 |
| OxfordPets | 37 | 5910 | 697 | 742 |
| Food101 | 101 | 75750 | 10100 | 15150 |
| EuroSAT | 10 | 18900 | 5400 | 2700 |
| 102flowers | 102 | 6552 | 818 | 819 |
| Describable Textures Datasets | 47 | 1880 | 1880 | 1880 |

Table 4: Dataset partition of the train, validation , and test set of total nine datasets we used.

