# OpenReview forum: "Data Descriptions from Large Language Models with Influence Estimation"
_ICLR.cc/2024/Conference — ICLR 2024 Conference Withdrawn Submission_

### Official Review · Reviewer_gMvU · 2023-10-30

**Soundness:** 2 fair
**Presentation:** 1 poor
**Contribution:** 2 fair
**Rating:** 3
**Confidence:** 5

**Summary:**

This paper builds upon the work of [1], which modifies CLIP's zero-shot classification by replacing class names with attributes generated by GPT. For instance, instead of using the class name "hen," [1] calculates the average cosine similarity between the image and a list of attributes like "two legs," "red, brown, or white feathers," "a small body," etc. This method proved more effective than using the class name alone. The current paper advances this concept in three main ways:

1. Attribute Generation: It enhances attribute generation by incorporating external knowledge from Wikipedia, unlike [1], which relies solely on GPT's internal knowledge. By including a Wikipedia URL in the prompt, the paper enables GPT to generate more accurate class descriptions.

2. Selection of Descriptions: The paper introduces scoring functions to identify the most informative class descriptions, termed "proponent texts." For instance, "red, brown, or white feathers" might be selected as the most defining feature of the class "hen" from the list of attributes.

3. Classification Methodology: Instead of direct zero-shot classification, the paper proposes training a linear classifier atop CLIP image embeddings, followed by further training using CLIP text embeddings derived from these texts. This approach is based on [2], which shows that image and text embeddings are equivalent in CLIP space, with text embeddings serving as "pseudo-images".

The authors demonstrate that on 7 out of 9 datasets, their method for obtaining text descriptions (Steps 1 and 2) outperforms [1] in zero-shot classification. Additionally, on all 9 datasets, Step 3 enhances supervised classification performance compared to models trained solely on images.

Reference:

[1] Visual Classification via Description from Large Language Models. Sachit Menon, Carl Vondrick. ICLR 2023. https://arxiv.org/abs/2210.07183

[2] Diagnosing and Rectifying Vision Models using Language. Yuhui Zhang, Jeff Z. HaoChen, Shih-Cheng Huang, Kuan-Chieh Wang, James Zou, Serena Yeung. ICLR 2023. https://arxiv.org/abs/2302.04269

**Strengths:**

- Incorporating external knowledge in Step 1 and selecting the most informative class description using scoring functions in Step 2 are innovative approaches that could offer deeper insights into model predictions.

- Experimental results show improvements in model performance when the proposed method is used, thereby demonstrating its practical value.

**Weaknesses:**

- The paper is challenging to follow, particularly the underlying motivation. I have to read many times to understand the motivation, methods, and results (see my summary). It requires significant revision for clarity before being published.

- The improvements over the [1] baseline are modest. In the most critical comparison of "Baseline (GPT-3.5)" versus "Ours Zero-shot" (Table 1), which compares the quality of class descriptions, the gains on 7 out of 9 datasets are relatively minor (typically 1-3%).

- The paper needs more in-depth analysis. Section 4.4 and Table 3 indicate a human preference for the class descriptions generated by this paper over [1], but the reasons for this preference remain unexplored. Given the marginal zero-shot improvement, a more thorough analysis of these descriptions could provide additional insights.

- The paper needs more quantitative analysis. Section 4.5 and Figure 6 qualitatively suggest that their class descriptions can reduce bias in text-to-image generation (e.g., for "doctors"), which is interesting but lacks numerical backing. Also, it's unclear why their class descriptions specifically mitigate gender bias in this context.

**Questions:**

- Equation 1 and Figure 4: How does the influence score based solely on images help select the most informative texts? Figure 4 shows a uniform influence of 116.924 on all the texts when computing the "Influence Score", isn't that expected?

- Table 3: The "%" symbol should be removed, or each number should be multiplied by 100.

---

### Official Review · Reviewer_vMa1 · 2023-11-01

**Soundness:** 3 good
**Presentation:** 2 fair
**Contribution:** 2 fair
**Rating:** 3
**Confidence:** 3

**Summary:**

The authors introduce a new method to comprehend and extract information that best explains each class within a dataset. They achieve this by integrating knowledge from external knowledge bases accessed through large language models like GPT-3.5. authors utilize influence estimation, combined with the CLIP score, to select the most pertinent textual descriptions. These descriptions can shed light on the model's focal points during predictions. Additionally, by leveraging recent advancements in vision-language contrastive learning for cross-modal transferability, the authors introduce a new benchmark task: cross-modal transfer classification. This is done to evaluate the potency of the data description. The effectiveness of the framework is demonstrated through extensive experiments on nine datasets.

**Strengths:**

- Multiple datasets and baselines are comprehensively compared in evaluating the effectiveness of the proposed framework.
- The proposed method is straightforward, intuitive, and easy-to-follow. Readers can easily understand the purposes and follow the content of the paper.

**Weaknesses:**

- The hallucination of GPT-3.5 can undermine the credibility of explanations from the suggested framework. The paper overlooks the topic of countering the effects of hallucination, rendering the paper less persuasive. Although the paper claims to use external knowledge to mitigate the misinformation aspects, there's a lack of discussion on how to deal with implicit hallucinations, which are even more challenging than preventing explicit hallucinations.
- The entire framework appears to pipe various existing methods, with the authors only utilizing GPT-3.5 as the conduit to integrate all these approaches into their system.
- The manuscript primarily highlights a few case studies addressing model bias, which renders the assessment somewhat unconvincing, especially since the paper claims to mitigate the algorithmic bias. The authors are highly encouraged to provide empirical experiments on bias mitigation. Otherwise, the contribution may not be promising.
- (Minor) The paper's writing could be enhanced, as several grammatical errors are evident in the paragraph. Some sentence structures make it hard for readers to understand the meanings.

**Questions:**

- According to the abstract, the author's use of the word "may" suggests there's inherent uncertainty in their method on providing interpretability and bias mitigation. Does this imply that the proposed method could potentially fail to provide explanations or sufficiently address biases at times?

---

### Official Review · Reviewer_ANzd · 2023-11-02

**Soundness:** 2 fair
**Presentation:** 1 poor
**Contribution:** 2 fair
**Rating:** 1
**Confidence:** 5

**Summary:**

The paper's propose a modification over Menon & Vondrick (2003) where they use external database (wikipedia) along with LLM (GPT3.5) to suggest relevant concepts for image classification. They show that their method achieves better zero shot performance over prior work when using CLIP embeddings. Furthermore, the authors propose a metric called Influence scores for text (IFT) to find which texts are highly influential for predicting a particular image is of a particular class. Using this metric they find important text (called proponents) for each class, which is used to improve performance of classifiers for image classification tasks.

**Strengths:**

I think the idea of using text to enhance classification performance of images is intriguing.

**Weaknesses:**

I find the paper poorly written and contributions marginal relative to prior work. Thus, I recommend rejection of the manuscript in its current form. Detailed reasons in questions.

**Questions:**

I think the positioning of the paper is not clear and contributions not well-motivated.

1. The authors first contribution is to extent prior work on using LLM for specifying textual concepts for classification by augmenting the LLMs with external knowledge bases. However, the way the authors achieve this is by simply supplying a wikipedia link of the classes as prompts to the LLM. I find this of limited novelty relative to prior work (Menon & Vondrick).

2. The authors second contribution is to propose an influence score metric for different text. I find it hard to follow the writing in this section (3.2). Following are some of the main points.
 * TracIn was proposed to measure the influence of some training data point on some test datapoint through how the loss function changes over training. However, if the authors are simply adding the influence of the entire training data on the entire validation set, then why use the approximation (involving dot products of gradients) at all? According to Lemma 3.1 in Pruthi et al 2020, isn't this just the difference between the loss functions evaluated for the entire validation set before and after training?
* The way equation 2 is written it seems the IFT for any text has the same first term making that redundant. However, the text says that the influence scores are used to select proponent images and then CLIP scores are used to select proponent text. It is not clear from the text of equation 2 how proponent images are selected? Also, if the text is extracted per class (as described in section 3.1) what does it mean to define IFT for a text by averaging it's influence scores over every class?

3. The last contribution of the authors is on some cross-modal transfer classification. It seems to be this is simply a post-hoc weighing of the loss function. The task is never described clearly in the paper. Is the goal to classify the class based on text inputs? or is it still an image classifier but just improved using further textual cues? What is the exact setup here? Algorithm 1 in the appendix is not well-written. I think readability would be improved if each separate aspect of the pipeline is well differentiated in the algorithm .

---

### Official Review · Reviewer_TWbL · 2023-11-06

**Soundness:** 2 fair
**Presentation:** 1 poor
**Contribution:** 1 poor
**Rating:** 3
**Confidence:** 5

**Summary:**

This paper proposes an approach to explain training data most relevant to a classification model through text descriptions of image classes from an LLM (GPT-3.5) by using a two-stage prompting process referencing an external knowledge base (Wikipedia). It aims to provide human-interpretable explanations for classification decisions by scoring text descriptions with a combination of influence functions and CLIP scores. The authors propose cross-modal transfer classification as a novel benchmark task and evaluate performance on nine datasets where they outperform their baseline of choice.

**Strengths:**

- Good coverage of datasets - both commonly used for classification tasks, and more niche ones - in a variety of domains.

- Thorough re-implementation of the baseline (Menon & Vondrick), with both reported performance, reproduced performance, and performance of the baseline method using GPT-3.5 instead of the original GPT-3 for a fair comparison.

- Interesting problem and relevant to practitioners of computer vision for a variety of tasks at scale: labelling new datasets, object classification at scale, classification on a fusion of datasets from various domains, addressing representational bias, and explainability.

**Weaknesses:**

- Paper is generally poorly structured and has some grammatical errors that make the reading flow difficult. It would be helpful if the abstract was shorter and to the point, if the introduction was clearer on the problem the paper addresses and why it is important, and if the experiment section had small introductions to each subsection to assist the flow.

- The two-step process for extracting textual descriptions with GPT-3.5 seems difficult to reproduce: the first step is to get the components of each class from the raw class name, and second step is to get the descriptions of each class using those components while providing a Wikipedia URL. Is that sufficient to address most commonly used datasets' label sets? What happens when working with open-vocabulary settings where the model encounters unseen concepts?

- The human evaluation can be redesigned to better quantify the metrics the authors are interested in: eg. "relevant", "helpful", and "informative" are subjective concepts and cannot be quantified across different participants in the user study without some reference. I suggest the authors take a look at user studies design with the goal of solving a task, where the human performance on that auxiliary task itself would be the metric for helpfulness or informativeness of the text description. An example of this can be found in: Plummer, B.A., Vasileva, M.I., Petsiuk, V., Saenko, K., Forsyth, D.: Why do these match? Explaining the behavior of image similarity models. (ECCV 2020). As the metrics reported in Section 4.4, Table 3 currently stand, I can't quite place in context the performance difference with the baseline (Menon & Vondrick 2023).

- Section 4.5 touches on a separate problem: model bias in generative models, which may be best served by a followup submission and requires a lot deeper dive into current bias evaluation and mitigation approaches from the fairness literature. It sits a bit estranged from the rest of the paper, and the claim that this methodology can "solve" the bias problems of generative models is a bit bold if it's only substantiated by one qualitative example. Typically, a sample of images with the same prompt need to be generated repeatedly such that a reliable estimate of how often sensitive concepts appear can be formed (eg. "female doctor"). A human evaluator pool is also often required to confirm the presence of the desired demographic attributes (eg. annotate the gender of the subject in the generated images). While I understand the proof-of-concept of how this methodology can assist in steering prompting in text-to-image generative models to a more "unbiased" sample, this is a complex problem and thorough experimentation would be required to make this point. I encourage the authors to also think carefully about the differences between "de-biasing" a generative model's outputs, and artificially creating a distribution that does not match the end user's expectation of such models.

**Questions:**

Suggested add to related work: Plummer, B.A., Vasileva, M.I., Petsiuk, V., Saenko, K., Forsyth, D.: Why do these match? Explaining the behavior of image similarity models. (ECCV 2020). Relevant because of the goal of producing human-interpretable explanations for vision-language retrieval tasks, and because of the user study design that might be helpful.